# Efficient Low-rank Backpropagation for Vision Transformer Adaptation

**Yuedong Yang   Hung-Yueh Chiang   Guihong Li   Diana Marculescu   Radu Marculescu**
Chandra Family Department of Electrical and Computer Engineering
The University of Texas at Austin
`{albertyoung,hungyueh.chiang,lgh,dianam,radum}@utexas.edu`

## Abstract

The increasing scale of vision transformers (ViT) has made the efficient fine-tuning of these large models for specific needs a significant challenge in various applications. This issue originates from the computationally demanding matrix multiplications required during the backpropagation process through linear layers in ViT. In this paper, we tackle this problem by proposing a new Low-rank Back-Propagation via Walsh-Hadamard Transformation (LBP-WHT) method. Intuitively, LBP-WHT projects the gradient into a low-rank space and carries out backpropagation. This approach substantially reduces the computation needed for adapting ViT, as matrix multiplication in the low-rank space is far less resource-intensive. We conduct extensive experiments with different models (ViT, hybrid convolution-ViT model) on multiple datasets to demonstrate the effectiveness of our method. For instance, when adapting an EfficientFormer-L1 model on CIFAR100, our LBP-WHT achieves 10.4% higher accuracy than the state-of-the-art baseline, while requiring 9 MFLOPs less computation. As the first work to accelerate ViT adaptation with low-rank backpropagation, our LBP-WHT method is complementary to many prior efforts and can be combined with them for better performance. Code: `https://github.com/SLDGroup/LBP-WHT`

## 1  Introduction

Vision transformers (ViT) have emerged as the latest state-of-the-art tool in numerous general computer vision tasks [1–7]. However, tailoring these models to meet specific needs (*e.g.*, new dataset with different distribution) can be challenging. Indeed, adapting ViT models via finetuning demands considerable computational resources and is often impractical for most edge applications. For instance, to maintain privacy, in federated learning [8–10], model adaptation is limited to users' personal edge devices (*e.g.*, smartphones), where computational power is tightly restricted [11, 12].

The primary computational bottleneck arises from gradient propagation through the dense layers of ViT. Specifically, calculating gradients for layer weights and inputs requires two computationally-intensive matrix multiplications, given the gradient for output [13]. To tackle this issue, [14] tries to simplify matrix multiplications using low-rank reparametrization. However, this method only reduces the gradient computation for weights and **not** for inputs, thus limiting the overall speedup. This observation raises the following question:

*How can we decrease the computational cost for all operations, including gradient computations for weights and inputs, involved in backpropagation (BP) through any linear layer in the ViT model?*

To answer this question, we introduce a new ***Low-rank BackPropagation via Walsh-Hadamard Transformation*** (LBP-WHT) method. As shown in Figure 1, our method intuitively performs BP for gradients w.r.t. inputs and weights in a low-rank space. To achieve this, we project the gradient w.r.t.

37th Conference on Neural Information Processing Systems (NeurIPS 2023).

the output into a low-rank space using WHT [15], then perform low-rank matrix multiplications, and finally project the results back. This way, all matrix multiplications occur in a low-rank space, hence the computational cost is significantly reduced. In summary, our contributions are as follows:

- We propose LBP-WHT, a new approach which greatly reduces the computational cost for adapting ViT while maintaining accuracy; our method lowers the computational barrier and enables adapting large ViT models on resource constrained edge devices.

- LBP-WHT is the first work accelerating ViT training by low-rank BP; thus, LBP-WHT is orthogonal to prior works and can be combined with them for a better performance. Additionally, LBP-WHT offers abundant flexibility that can provide a good tradeoff between accuracy and cost.

- Extensive experiments on multiple datasets demonstrate the effectiveness of our method. Indeed, LBP-WHT consistently outperforms the baseline methods both in accuracy and speed. For instance, LBP-WHT achieves 10.4% higher accuracy, while requiring 9 MFLOPs less computation than [14] for training EfficientFormer-L1 on CIFAR100 dataset.

The paper is organized as follows. Section 2 formulates the problem associated with BP for linear layers. Section 3 presents our method LBP-WHT in detail. Experimental results are presented in Section 4. Section 5 reviews relevant work. Finally, Section 6 summarizes our main contributions.

## 2   Problem Formulation

**Naming conventions:** In this paper, we treat all feature maps as matrices composed of real numbers, with dimensions $\mathbb{R}^{C \times L}$, where $C$ represents the number of rows and $L$ denotes the number of columns. Each row in the matrix is regarded as a "channel" consisting of $L$ elements, and there are a total of $C$ channels in the feature map. We use subscripts to identify specific variables, such as $C_x$ for the number of channels associated with variable $x$. Gradients with respect to $x$ are denoted by $g_x$, with the subscript indicating the target variable $x$.

**Backpropagation for linear layers:** We focus on the BP process for linear layers, a crucial building block for vision transformers. Given an input $x \in \mathbb{R}^{C_x \times L}$ and weights $w \in \mathbb{R}^{C_y \times C_x}$, the forward propagation to compute the output $y \in \mathbb{R}^{C_y \times L}$ can be expressed as:

$$y = x \cdot w^T \tag{1}$$

Therefore, as shown in Figure 2a, given the gradient with respect to the output $y$, *i.e.*, $g_y \in \mathbb{R}^{C_y \times L}$, the back-propagation for computing the gradient with respect to the weights $w$, $g_w \in \mathbb{R}^{C_y \times C_x}$, and the gradient with respect to the input $x$, $g_x \in \mathbb{R}^{C_x \times L}$, can be represented as two matrix multiplications:

$$g_w = g_y \cdot x, g_x = g_y \cdot w \tag{2}$$

The gradient w.r.t. the weight ($g_w$) is utilized for updating the weights $w$, while the gradient w.r.t. the input ($g_x$) is employed for propagating the gradient to other layers. During the BP process, each matrix multiplication incurs a computational cost of $2C_xC_yL$ FLOPs, which amounts to $4C_xC_yL$ FLOPs, in total. Given that in ViT models, the number of channels ($C_x$ and $C_y$) and the length of the input feature map ($L$) are substantial [1–7], the computational cost for BP becomes significant.

**Low-rank backpropagation:** As shown in Figure 1 and 2b, we propose reducing the computational cost for both matrix multiplications by employing low-rank approximations. Specifically, we first project variables into a low-rank space as follows:

$$\hat{g}_y = p(g_y), \hat{x} = p(x) \tag{3}$$

Here, $\hat{g}_y \in \mathbb{R}^{C_y \times R}$ and $\hat{x} \in \mathbb{R}^{C_x \times R}$ represent the low-rank space projections ($R << L$) for the gradient with respect to the output ($g_y$) and input $x$, respectively. The projection function $p(\cdot)$ will be introduced in the next section.

Next, we execute the BP through the linear layer in the low-rank spaces as follows:

$$\hat{g}_w = \hat{g}_y \cdot \hat{x}, \hat{g}_x = \hat{g}_y \cdot w \tag{4}$$

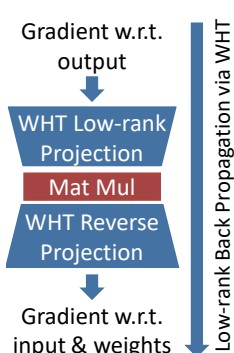

Figure 1: Our LBP-WHT. "Mat Mul" is short for "Matrix Multiplication".

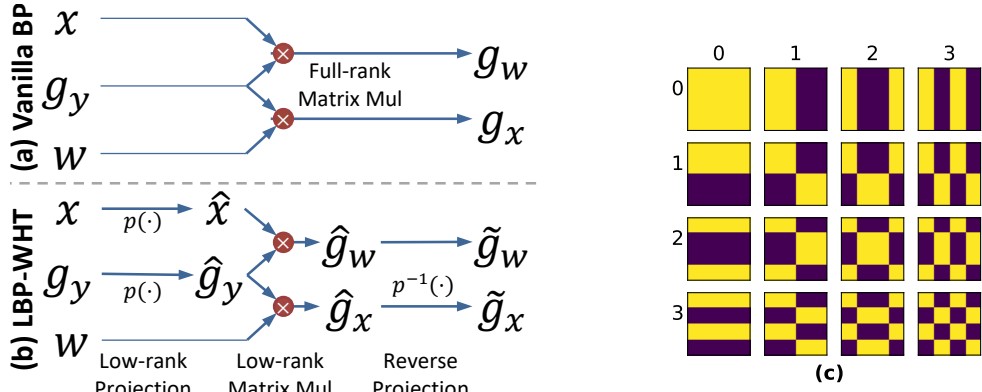

Figure 2: **(a-b)** Workflows for BP through a linear layer utilizing (a) the conventional method and (b) our LBP-WHT method. The intuition is to reduce the computation cost for BP by performing matrix multiplication in a low-rank space. To achieve this, we first project variables into a low-rank space using WHT $p(\cdot)$, then carry out efficient matrix multiplications, and finally project them black using $p^{-1}(\cdot)$, where both $p$ and $p^{-1}$ are implemented with WHT. **(c)** Bases $B_{i,j}$ for order-4 2D WHT. White and Black represent +1 and -1, respectively. Of note, in the context of ViT, 2D feature maps are flattened into 1D, so we utilize a flattened version of these bases.

Finally, we project the low-rank gradient with respect to the input ($\hat{g}_x$) back into its original space. The reverse projection for $\hat{g}_w$ can be omitted as it already exists in the same space $\mathbb{R}^{C_y \times C_x}$ as the target $g_w$. For $\hat{g}_x$, the reverse projection is accomplished using the function $p^{-1}(\cdot)$, the details of which will be presented later:

$$\tilde{g}_w = \hat{g}_w, \tilde{g}_x = p^{-1}(\hat{g}_x) \tag{5}$$

Here, $\tilde{g}_w$ and $\tilde{g}_x$ represent the resulting gradients for weights and input. As these gradients are generated through an approximated back-propagation process rather than the standard BP, we denote these variables with tildes.

## 3    LBP-WHT: Low-rank BackPropagation via WHT

As shown in Figure 2b, intuitively, we reduce the computational cost by performing back-propagation in a low-rank space, as described in Equation 4. For instance, using a rank $R$ approximation, each matrix multiplication requires $2C_x C_y R$ FLOPs, which can be substantially smaller than $2C_x C_y L$ when $R \ll L$. Nevertheless, this approach necessitates two additional steps, projection and reverse projection (as illustrated in Equation 3 and 5), which introduce some computational overhead. Furthermore, the low-rank projection may add noise and potentially diminish the quality of training. To address these concerns, our method incorporates a low-overhead projection function based on the WHT and tackles the second issue by selecting an appropriate set of WHT bases.

WHT is a generalized Fourier transformation. Figure 2c displays the transformation basis for an order-4 WHT. For an order-$n$ 2D WHT, there are $n \times n$ bases $B_{i,j}$, with each basis being an $n \times n$ matrix containing only +1 and −1. Of note, in the context of ViT, 2D feature maps are flattened into 1D maps, so we utilize a flattened WHT base—a vector with a length of $n^2$, *i.e.*, $B_{i,j} \in \mathbb{Z}^{n^2 \times 1}, 0 \leq i, j < n$. WHT possesses four properties that make it advantageous for us:

- The transformation bases are complete.
- The transformation bases are orthogonal.
- The transformation bases contain only +1 and −1.
- The transformation cost can be reduced via fast WHT algorithm with $O(n \log n)$ complexity.

The first property (completeness) allows WHT to perform transformations ranging from lossy (when few bases are activated) to lossless (when all bases are activated). This grants flexibility in exploring the trade-off between efficiency and accuracy. The second property ensures that any variable has precisely one projection result, obtainable via matrix multiplication. For instance, the projection

function for $g_y$ (Equation 3) with basis $B_{i,j}$ can be expressed as $p(g_y) = g_y \cdot B_{i,j}$. Likewise, the reverse projection can also be implemented using a simple matrix multiplication. The third and final properties demonstrate the efficiency of WHT implementation, requiring only $O(n \log n)$ additions/subtractions and no multiplications [16].

## 3.1 Low-rank Back-Propagation with WHT

Indeed, these four properties demonstrate that WHT is an ideal fit for our needs, offering both low overhead and high flexibility for selecting an appropriate set of bases. Therefore, we employ WHT as the projection function $p(\cdot)$ and reverse projection function $p^{-1}(\cdot)$ in Equations 3 and 5. More specifically, for an order-$n$ WHT with a set of $R$ bases chosen by an index set $\mathcal{I}$, the projection function can be written as:

$$p(x) = \text{WHT}(x; \mathcal{I}) = x \cdot (B_{i_1,j_1} \quad B_{i_2,j_2} \quad \cdots B_{i_R,j_R}), (i_k, j_k) \in \mathcal{I}, 1 \le k \le R \tag{6}$$

where $\mathcal{I} = \{(i_k, j_k) | 1 \le i_k, j_k \le n, 1 \le k \le R\}$ indicates which bases are activated. Similarly, the reverse projection function can be expressed as:

$$p^{-1}(x) = \text{WHT}^{-1}(x; \mathcal{I}) = x \cdot (B_{i_1,j_1} \quad B_{i_2,j_2} \quad \cdots B_{i_R,j_R})^T, (i_k, j_k) \in \mathcal{I}, 1 \le k \le R \tag{7}$$

For simplicity, both Equations 6 and 7 are presented using the vanilla WHT algorithm with computational complexity $O(n^2)$, rather than the fast WHT algorithm with complexity $O(n \log n)$. Consequently, our LBP-WHT algorithm can be summarized as Algorithm 1 also shown in Figure 2b.

---

**Algorithm 1** Backpropagation through a linear layer with LBP-WHT.

---

**Input:** Input $x$, weight $w$, gradient w.r.t. output $g_y$, Selected WHT base indices $\mathcal{I}$
**Output:** Approximated gradient w.r.t. input $\tilde{g}_x$, approximated gradient w.r.t. weight $\tilde{g}_w$

$\hat{x} \leftarrow p(x) = \text{WHT}(x; \mathcal{I})$                $\triangleright$ Projection to a low-rank space with WHT (Equation 3)
$\hat{g}_y \leftarrow p(g_y) = \text{WHT}(g_y; \mathcal{I})$
$\hat{g}_w \leftarrow \hat{g}_y^T \cdot \hat{x}$              $\triangleright$ Efficient matrix multiplication in a low-rank space (Equation 4)
$\hat{g}_x \leftarrow \hat{g}_y \cdot w$
$\tilde{g}_x \leftarrow p^{-1}(\hat{g}_x) = \text{WHT}^{-1}(\hat{g}_x; \mathcal{I})$      $\triangleright$ Reverse projection to a full-rank space (Equation 5)
$\tilde{g}_w \leftarrow \hat{g}_w$          $\triangleright$ Skipped reverse projection since $\hat{g}_w$ is already in a full-rank space

---

Given input for BP, we first project $x$ and $g_y$ into low-rank space (Equation 3), then we performs matrix multiplication (Equation 4) and lastly we project the results back (Equation 5).

## 3.2 WHT Bases Selection

Here we explore two types of basis selection strategies: low-pass and low-heuristic-error.

**Low-pass (LP) Base Selection:** Natural images have strong spatial locality, *i.e.*, pronounced low-frequency components [17, 18]. We take advantage of this feature and choose bases with stronger low-frequency responses, which have smaller indices as illustrated in Figure 2c. More specifically, we consider both $L_1$-based and $L_\infty$-based low-pass basis selection strategies ($\text{LP}_{L_1}$ and $\text{LP}_{L_\infty}$):

$$\mathcal{I}_{L_1} = \{(i_k, j_k) \mid |i_k| + |j_k| \le r, \ 1 \le i_k, j_k \le n\}, \text{LP}_{L_1}\text{-}r \text{ selection} \tag{8}$$

$$\mathcal{I}_{L_\infty} = \{(i_k, j_k) \mid \max(i_k, j_k) \le r, \ 1 \le i_k, \ j_k \le n, \ \}, \text{LP}_{L_\infty}\text{-}r \text{ selection} \tag{9}$$

$\mathcal{I}_{L_1}$ and $\mathcal{I}_{L_\infty}$ are the index sets for selecting WHT bases, as described in Section 3.1. For example, with $\text{LP}_{L_1}$-2 base selection, three bases are chosen, *i.e.*, $\mathcal{I}_{L_1} = \{(0,0), (0,1), (1,0)\}$, and the rank for projection, namely $R$, is three.

**Low-heuristic-error (LHE) Base Selection:** According to Parseval's Theorem [19], WHT preserves the signal energy, so by selecting the WHT bases with the top-$r$ strongest responses, we can preserve most energy during low-rank projection and minimize the error. Since profiling the energy for all WHT bases on all training steps is expensive, we profile the energy for all WHT bases only for a small number of training steps and select the bases with the top-$R$ energy.

Considering that the $L_1$-based low-pass basis selection has a much lower profiling overhead than the low-heuristic-error basis selection and provides finer granularity in balancing accuracy and efficiency, we primarily focus on the $\text{LP}_{L_1}$ selection method and explore the other two in Section 4.5.

### 3.3 Overhead Analysis

Since the computational cost for the fast WHT algorithm depends on the basis selection, we simplify the analysis in this section by considering the matrix multiplication-based vanilla WHT algorithm, as shown in Equations 6 and 7. Table 1 presents the computation requirements for a linear layer with input and output channels $C_x$ and $C_y$, feature map size $L$, and the rank for low-rank WHT approximation $r$. Our LBP-WHT achieves a $\frac{L}{R}$ times speedup with an overhead of $(2C_x + C_y)LR$ FLOPs, which is only $\frac{(2C_x+C_y)LR}{4C_xC_yL}$ or $(\frac{1}{C_x} + \frac{1}{2C_y})\frac{R}{2}$ of the total computation required by vanilla BP. Given that ViT typically has a large number of channels, the overhead is very small.

|  | FLOPs |
|---|---|
| Vanilla BP | $4C_xC_yL$ |
| Projection | $(C_x + C_y)LR$ |
| Low-rank MM | $4C_xC_yR$ |
| Reverse Projection | $C_xLR$ |

Table 1: Computation required by vanilla BP and components in our LBP-WHT. We consider the projection and reverse projection as overhead. "MM" is short for "Matrix Multiplication".

For instance, the final linear layer in SwinV2-small [1] consists of 3072 input channels, 768 output channels, and a feature map size of 49, which means $C_x = 3072$, $C_y = 768$, and $L = 49$. As per Table 1, conventional backpropagation (BP) requires 462.3 MFLOPs. In contrast, our Low-Rank Backpropagation with WHT (LBP-WHT) method, assuming a rank of 8 ($R = 8$), needs only 78.2 MFLOPs, which is roughly 16.9% of the computation required by vanilla BP.

Breaking down the 78.2 MFLOPs for LBP-WHT, we see that 1.5 MFLOPs are needed for the low-rank projection, 75.5 MFLOPs for BP in the low-rank space, and 1.2 MFLOPs for the reverse projection. The combined overhead is 2.7 MFLOPs, accounting for just 0.6% of vanilla BP's computation and 3.5% of LBP-WHT's computation. This demonstrates that with WHT, we can significantly reduce the computation for BP while incurring negligible overhead for low-rank projection.

## 4 Experimental Results

In this section, we first present our experimental results on image classification and semantic segmentation tasks. Then, we explore the impact of different ranks for low-rank projection and different base selection strategies. Lastly, we present our preliminary results for deploying our methods on real edge devices in the supplementary material.

### 4.1 Experimental Setup

**Environment:** We setup our environment with PyTorch 1.13, MMClassification v0.25 and MMSegmentation v0.30. Models are trained with an NVIDIA-A6000 GPU.

**Classification:** We conduct experiments for image classification following [20]. We use ImageNet [21]-pretrained ViTs and finetune them on six different datasets, namely, CIFAR100 [22] (CF100), CIFAR10 [22] (CF10), Cars [23], Flowers [24], Food [25], and Pets [26]. We standardize the image resolution across all datasets to 224×224. Each model is finetuned for 50 epochs using the AdamW [27] optimizer and a batch size of 64. The learning rate is adjusted for each dataset based on the performance of EfficientFormer-L1 [28] with vanilla BP.

**Semantic Segmentation:** We use the ADE20K [29]-pretrained Segformer-mit-b0 [30] model and finetune it on two datasets, Cityscapes [31] (City) and the enhanced Pascal-VOC 2012 [32] (VOC12A). The images are downscaled and cropped to a size of 512×512 pixels for training. Models are finetuned for 20,000 steps using the AdamW optimizer and a batch size of 8.

**Partial Training:** We primarily report on the results of training the final stage of the ViT using various methods, a common approach in transfer learning to reduce the computational cost [18, 33–36]. More results for full training are included in the supplementary material.

**Baselines Comparisons:** We compare our results against three baseline methods: Full BP, "LoRA", and "LoRA-all". Full BP refers to training the model with standard full-rank backpropagation. "LoRA" and "LoRA-all" are methods derived from [14]. "LoRA" strictly follows [14], which uses low-rank reparametrization solely in the ViT's attention modules, while "LoRA-all" applies this method to all linear layers. For hybrid CNN-ViT models, where the attention modules are usually only in the final stage, we use "LoRA-all" for full training.

| | | | | | Partial Training: Training the Last Stage | | | | | | |
|---|---|---|---|---|---|---|---|---|---|---|---|
| **Model** | **Method** | **R** | **Speedup** | **mAcc** | **MFLOPs** | **CF100** | **CF10** | **Cars** | **Flowers** | **Food** | **Pets** |
| Efficient Former [28] L1 (Hybrid) | Full BP | - | 1.0× | 88.66 | 1685.01 | 79.28 | 95.23 | 84.80 | 95.50 | 84.04 | 93.13 |
| | LoRA | 8 | 6.9× | 79.59 | 242.61 | 65.25 | 87.40 | 65.76 | 90.16 | 76.46 | 92.50 |
| | LoRA-all | 8 | 1.7× | 85.97 | 976.50 | 76.92 | 94.38 | 76.84 | 93.56 | 81.50 | 92.64 |
| | $\text{LP}_{L_1}$-2★ | 3 | **7.2×** | **85.50** | 233.62 | 75.61 | 93.35 | 76.96 | 95.07 | 79.65 | 92.34 |
| | $\text{LP}_{L_1}$-4★ | 10 | **3.5×** | **87.76** | 480.00 | 78.27 | 94.60 | 82.60 | 95.53 | 82.37 | 93.16 |
| | $\text{LP}_{L_1}$-8 | 36 | 1.2× | **88.62** | 1397.02 | 79.34 | 95.31 | 84.57 | 95.58 | 83.98 | 92.94 |
| Efficient Former [28] L7 (Hybrid) | Full BP | - | 1.0× | 91.91 | 11071.73 | 86.40 | 97.61 | 87.48 | 97.19 | 88.58 | 94.22 |
| | LoRA | 8 | 2.0× | 88.45 | 5520.52 | 81.66 | 95.44 | 78.95 | 95.82 | 85.15 | 93.65 |
| | LoRA-all | 8 | 1.9× | 90.36 | 5973.40 | 85.09 | 97.10 | 83.66 | 96.16 | 86.60 | 93.54 |
| | $\text{LP}_{L_1}$-2★ | 3 | **9.2×** | **89.88** | 1202.83 | 83.78 | 96.73 | 83.02 | 96.55 | 85.38 | 93.84 |
| | $\text{LP}_{L_1}$-4★★ | 10 | **3.8×** | **91.16** | 2905.16 | 85.10 | 97.22 | 86.01 | 97.14 | 87.48 | 94.03 |
| | $\text{LP}_{L_1}$-8 | 36 | 1.2× | **91.80** | 9241.53 | 86.19 | 97.62 | 87.32 | 97.40 | 87.77 | 94.47 |
| Efficient FormerV2 [37] S0 (Hybrid) | Full BP | - | 1.0× | 84.27 | 454.64 | 72.37 | 92.63 | 75.90 | 92.73 | 81.44 | 90.52 |
| | LoRA | 8 | 2.2× | 74.42 | 206.29 | 60.74 | 84.89 | 52.99 | 86.47 | 72.23 | 89.18 |
| | LoRA-all | 8 | 1.5× | 78.94 | 313.19 | 65.51 | 88.95 | 63.49 | 88.94 | 76.88 | 89.89 |
| | $\text{LP}_{L_1}$-2★ | 3 | **4.5×** | **77.53** | 99.94 | 65.75 | 88.68 | 59.02 | 89.51 | 74.72 | 87.49 |
| | $\text{LP}_{L_1}$-4★★ | 10 | **2.7×** | **81.29** | 168.60 | 69.03 | 90.88 | 68.34 | 90.73 | 79.45 | 89.29 |
| | $\text{LP}_{L_1}$-8 | 36 | 1.1× | **83.78** | 405.84 | 71.90 | 92.29 | 74.31 | 92.60 | 81.07 | 90.52 |
| Efficient FormerV2 [37] L (Hybrid) | Full BP | - | 1.0× | 91.03 | 3605.86 | 82.26 | 96.13 | 88.78 | 96.80 | 87.63 | 94.60 |
| | LoRA | 8 | 2.5× | 84.74 | 1469.66 | 74.35 | 92.94 | 70.99 | 92.97 | 82.81 | 94.36 |
| | LoRA-all | 8 | 1.7× | 87.94 | 2092.47 | 78.97 | 94.99 | 80.39 | 94.32 | 84.94 | 94.03 |
| | $\text{LP}_{L_1}$-2★ | 3 | **6.8×** | **86.88** | 533.61 | 78.00 | 94.28 | 76.05 | 94.36 | 84.14 | 94.47 |
| | $\text{LP}_{L_1}$-4★★ | 10 | **3.3×** | **89.47** | 1088.06 | 80.15 | 95.54 | 84.64 | 95.97 | 85.85 | 94.66 |
| | $\text{LP}_{L_1}$-8 | 36 | 1.1× | **90.79** | 3150.95 | 82.24 | 96.02 | 87.34 | 96.68 | 87.41 | 95.07 |
| SwinV2 [1] Small (ViT) | Full BP | - | 1.0× | 90.62 | 3896.51 | 80.84 | 96.07 | 85.35 | 97.61 | 88.31 | 95.53 |
| | LoRA | 8 | 2.4× | 78.00 | 1600.19 | 68.50 | 89.62 | 54.15 | 83.77 | 79.81 | 92.15 |
| | LoRA-all | 8 | 2.0× | 84.86 | 1974.72 | 73.33 | 92.29 | 74.78 | 90.99 | 84.61 | 93.16 |
| | $\text{LP}_{L_1}$-2★★ | 3 | **3.3×** | **90.32** | 1166.98 | 80.23 | 95.65 | 85.30 | 97.50 | 88.06 | 95.15 |
| | $\text{LP}_{L_1}$-4★★ | 10 | **2.5×** | **90.43** | 1535.36 | 80.39 | 95.71 | 85.30 | 97.54 | 88.32 | 95.34 |
| | $\text{LP}_{L_1}$-8 | 36 | 1.3× | **90.60** | 2932.84 | 80.80 | 95.80 | 85.72 | 97.56 | 88.19 | 95.53 |
| | | | | | Full Training | | | | | | |
| Efficient FormerV2 [37] S0 (Hybrid) | Full BP | - | 1.0× | 89.19 | 2259.93 | 84.06 | 96.88 | 84.80 | 93.62 | 84.99 | 90.79 |
| | LoRA-all | 8 | 1.2× | 86.07 | 1899.99 | 81.14 | 96.27 | 76.25 | 90.60 | 81.88 | 90.27 |
| | $\text{LP}_{L_1}$-4 | 10 | **1.9×** | 78.56 | 1186.67 | 72.93 | 92.67 | 51.14 | 90.68 | 74.62 | 89.34 |
| | $\text{LP}_{L_1}$-7★★ | 28 | **1.2×** | **87.86** | 1833.31 | 83.14 | 96.53 | 80.69 | 92.21 | 83.76 | 90.84 |
| | $\text{LP}_{L_1}$-8 | 36 | 1.1× | **88.56** | 2116.41 | 83.42 | 96.76 | 83.00 | 92.75 | 84.27 | 91.14 |

Table 2: Results for image classification. "$\text{LP}_{L_1}$-$r$" refers to our LBP-WHT method with $\text{LP}_{L_1}$-$r$ base selection as outlined in Equation 8. "mAcc" represents the mean accuracy across all datasets. "R" is short for "rank". "Hybrid" represents CNN-ViT-hybrid architecture. Results outperforming both LoRA and LoRA-all in speed and mAcc are underlined and marked with ★. Those exceeding all LoRA methods get ★★. Results that have higher speed or mAcc are highlighted in bold. More results are included in the supplementary material.

**Computation Measurements and Preliminary Deployment Results:** To determine the computational requirements of different models and methods, we run model training on an Intel 11900K CPU and measure the exact FLOPs using the embedded performance tools "perf" in the Linux kernel v5.15.87. For preliminary deployment results, we test our method on the last two linear layers of EfficientFormer-L1, using OpenBLAS and CuBLAS for CPU and GPU testing respectively on an NVIDIA Jetson Nano. The results for deployment are reported in the supplementary material.

## 4.2 Image Classification Results

Table 2 demonstrates the effectiveness of our LBP-WHT method in adapting ViT for image classification tasks. Here are some more specific observations:

**Comparison with LoRA-based baselines:** Our LBP-WHT method consistently surpasses the LoRA-based method across all eight datasets in both partial and full training modes. For instance, when only training the final stage of EfficientFormer-L1, LBP-WHT using $\text{LP}_{L_1}$-2 base selection requires **8.9 MFLOPs fewer computations** than LoRA, yet achieves **10% greater accuracy** on the CIFAR100 dataset. When the entire model is trained, the accuracy difference is smaller, but LBP-WHT still outperforms the LoRA-based method. For instance, in comparison to LoRA-all, LBP-WHT using $\text{LP}_{L_1}$-7 base selection requires less computation (66.68 MFLOPs), but still improves accuracy by 2% on CIFAR100 when training the EfficientFormerV2-S0 model.

**Comparison with traditional full-rank BP:** With $\text{LP}_{L_1}$-8 base selection, our LBP-WHT method either matches or surpasses the accuracy of full-rank BP while only requiring about 80% of the total

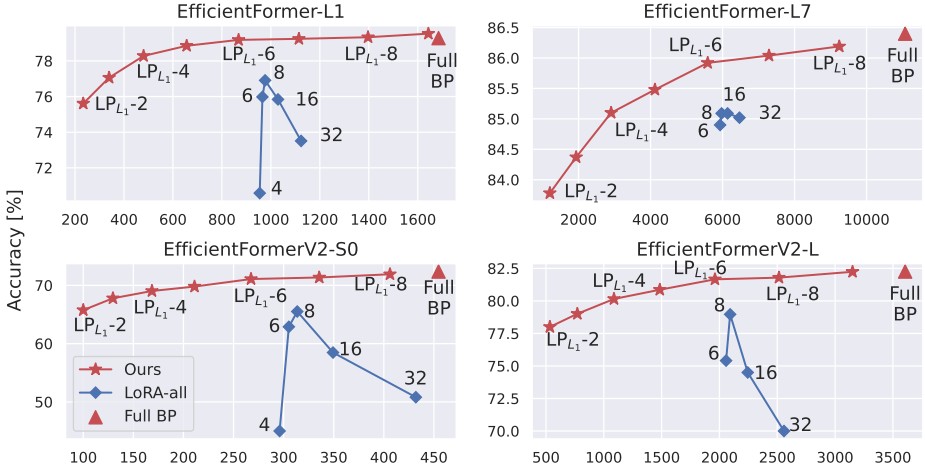

Figure 3: Accuracy and computation for training the last stage of different models with different ranks on CIFAR100 dataset. Our method consistently outperforms the baseline LoRA-all.

| Partial Training: Training Last Stage + Decoder | | | | | Full Training | | | |
|---|---|---|---|---|---|---|---|---|
| Method | R | MFLOPs | City | VOC12A | Method | R | MFLOPs | City | VOC12A |
| Full BP | - | 10052.00 | 62.85 | 69.30 | Full BP | - | 16700.26 | 67.37 | 70.84 |
| LoRA | 8 | 5854.61 | 51.43 | 58.18 | LoRA | 8 | 11976.46 | 62.57 | 58.18 |
| LoRA-all | 8 | 6262.01 | 58.07 | 66.26 | LoRA-all | 8 | 11971.13 | 65.74 | 67.82 |
| $LP_{L_1}$-2★ | 3 | **1481.94** | **58.95** | **67.93** | $LP_{L_1}$-2 | 3 | **5746.54** | 61.57 | **67.93** |
| $LP_{L_1}$-4★ | 10 | **2725.39** | **60.97** | **68.85** | $LP_{L_1}$-4★ | 10 | **7295.52** | **64.72** | **68.85** |
| $LP_{L_1}$-8 | 36 | 7308.45 | **62.68** | **68.95** | $LP_{L_1}$-8 | 36 | 13086.06 | **66.17** | **68.95** |

Table 3: Experimental results for semantic segmentation. Results are highlighted as in Table2.

computation. When using smaller ranks, LBP-WHT significantly reduces the cost with only minor accuracy costs. For example, when training the final stage of EfficientFormer-L1 using $LP_{L_1}$-4 base selection, LBP-WHT achieves a $3.5\times$ speedup with just a 1% loss in accuracy on CIFAR100. With $LP_{L_1}$-8 base selection, LBP-WHT achieves even **higher accuracy** (79.34%) with a $1.2\times$ **speedup**.

These results underscore the merits of our method. As shown in Table 2, our method achieves computational savings by systematically reducing the computational cost for **all** operations during backpropagation, including the gradient computation for both input and weight. Specifically, when we apply a similar rank for LoRA-all and LBP-WHT, we anticipate that both methods will have similar computational costs for computing the weight gradient. However, as LoRA-all *cannot* speed up the gradient computation for the input while LBP-WHT can, our LBP-WHT method requires only half the total computation of LoRA-all. Consequently, for a similar computational budget, LBP-WHT can employ a higher rank for low-rank projection, thus leading to a higher accuracy. For example, when training the entire EfficientFormerV2-S0 model, LBP-WHT with $LP_{L_1}$-4 (rank 10) only requires 1187 MFLOPs, which is 62% of the computational cost for LoRA-all. Thus, for a similar budget, LBP-WHT can use a rank 28 projection ($LP_{L_1}$-7) and achieve a higher accuracy.

### 4.3 Semantic Segmentation

Table 3 presents the experimental results for adapting the ADE20K-pretrained Segformer model on Cityscapes and augmented Pascal VOC 2012 dataset. Our LBP-WHT has better results in most cases. For instance, when partially training on the Cityscapes dataset, our approach using $LP_{L_1}$-4 base selection achieves a mIoU score approximately 0.9% higher than that of LoRA-all. Moreover, it only requires 1481.9 MFLOPs, which is $4.2\times$ faster. These findings not only further validate the efficacy of our method, but also demonstrate its broad applicability across key computer vision tasks.

### 4.4 Exploration 1: Different Ranks for Low-rank Projection in LBP-WHT

Figure 3 shows the accuracy achieved when adapting ImageNet-pretrained ViTs for CIFAR100, with varying ranks for low-rank model adaptation. Our observations from this figure are as follows:

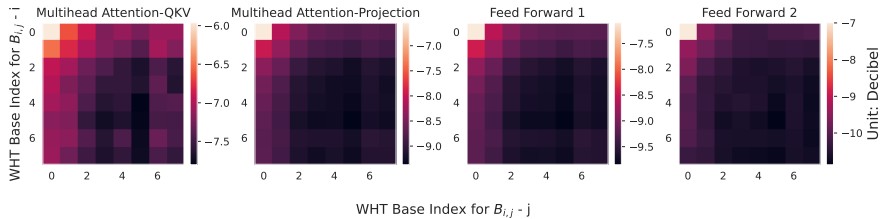

Figure 5: The WHT spectrum for gradient w.r.t. layer output ($g_y$) collected from the last attention block of EfficientFormer-L1. The brightness for each pixel $(i, j)$ in the spectrum represents the energy preserved by the WHT base $B_{i,j}$ during projection. A brighter pixel means a larger energy. As shown in Figure 2c, the WHT base with smaller indices corresponds to a lower frequency component.

1. Our LBP-WHT method consistently outperforms the LoRA-all method, *i.e.*, for a similar level of computation, LBP-WHT yields higher accuracy.

2. By altering the rank, LBP-WHT provides a broader range of cost options than the baseline method.

3. LBP-WHT's accuracy monotonically improves as more ranks are employed for projection.

4. For all models with our LBP-WHT method, a generally concave accuracy-computation curve is observed. This indicates strong diminishing returns in using larger ranks.

5. LBP-WHT with $LP_{L_1}$-6 base selection achieves an accuracy very close to that of full BP.

Our first observation further confirms the superior performance of our method. The second observation indicates the broad applicability of our method. For instance, for edge devices with limited computational budgets, like Raspberry Pi, we can employ LBP-WHT with a lower rank to reduce computational cost. On the other hand, for more powerful devices, such as personal desktops equipped with GPUs, a larger rank can be chosen to enhance accuracy. This ensures that users with various computational resources and constraints can benefit from our method.

The last three observations offer guidelines for rank selection with our LBP-WHT method.

**With strict computational constraints:** Given our third observation above, if there is a hard limit on the maximum number of FLOPs allocated for training, selecting the rank for LBP-WHT is straightforward: we simply opt for the largest possible number of ranks, which in most cases yields the highest accuracy.

**Without strict computational constraints:** Our final two observations suggest that training efficiency can be characterized by the marginal accuracy, or the slope of the accuracy-computation curve. As shown in Figure 4, before $LP_{L_1}$-4, the marginal accuracy is significantly greater than zero. However, after $LP_{L_1}$-6, the marginal accuracy is very close to zero. This implies that choosing fewer ranks than $LP_{L_1}$-4 or more than $LP_{L_1}$-6 may not be advantageous, as it could either forgo the opportunity for good performance with a small amount of computation or waste computation with little benefit. Thus, a good selection empirically lies between $LP_{L_1}$-4 and 6.

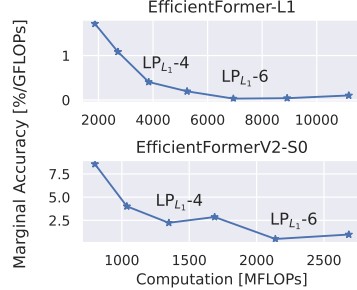

Figure 4: Marginal accuracy: the slope of the accuracy-computation curve in Figure 3.

### 4.5 Exploration 2: Different Bases Selection Method

Figure 5 shows the WHT spectrum for the gradient w.r.t. layer output ($g_y$) collected from the last attention block in EfficientFormer-L1. We observe that most energy concentrates in the low-frequency area, *i.e.*, the top-left corner, which supports claim in Section 3.2 that natural images have strong spatial locality and strong low-frequency components. Furthermore, Figure 5 demonstrates that by choosing WHT

| Method | Rank | CF100 | CF10 |
|---|---|---|---|
| $LP_{L_1}$ | 10 | 78.27 | 94.60 |
| $LP_{L_\infty}$ | 9 | 77.64 | 94.30 |
| LHE | 10 | 78.06 | 94.60 |

Table 4: Experimental results for adapting EfficientFormer-L1 on CIFAR100 and CIFAR10 with different base selection methods. Accuracy is in percentages (%)

bases with low-frequency responses - that is, using the selection methods $LP_{L_1}$ and $LP_{L_\infty}$ - we can preserve most of the energy and minimize error during the low-rank projection. As indicated in Table 4, both of these low-pass base selection methods yield accuracy levels very similar to those achieved with the low-heuristic-error (LHE) method. The LHE method profiles the WHT spectrum

and selects the WHT bases with the strongest response. Given that the $\text{LP}_{L_1}$ base selection method eliminates the need for profiling (unlike LHE) and offers a more favorable balance between accuracy and cost compared to $\text{LP}_{L_\infty}$, we have selected $\text{LP}_{L_1}$ as the standard method for LBP-WHT.

### 4.6 Limitation and Broader Impacts

**Full training with a small number of ranks:** As shown in Table 2, we find that the accuracy degradation is not negligible when using a small number of ranks for LBP-WHT full training. We consider this is the issue of accumulating error introduced by low-rank projection during BP. We expect that an improved approximation method can perform even better. Of note, even with accuracy degradation, our method still consistently outperforms the baselines, *i.e.*, LoRA-based methods.

**Broader Impact:** Our method greatly reduced the barrier for training ViTs. As a positive feature, our method may push the development of privacy-centric on-device training methods like federated learning; our method may also enable more researchers to test their ideas with powerful ViTs. On the other hand, our method may lower the barrier for irresponsible customization and use of ViT.

## 5  Related Work

**Low-rank Model Adaptation:** [14] proposes to speed up transformer training by attaching and training only low-rank branches to the linear layer. More precisely, consider a linear layer with equation $y = x \cdot w^T$, where $x$ is the input, $y$ is the output, and $w$ is the weight. LoRA adds a branch that contains low-rank weights $w_A$ and $w_B$, forming $y_{\text{LoRA}} = x \cdot w^T + x \cdot (w_A \cdot w_B)^T$. The original weight $w$ is kept frozen, while the appended weights $w_A$ and $w_B$ are trained. Since the ranks of $w_A$ and $w_B$ are much smaller than that of $w$, the computation needed to calculate the gradients with respect to $w_A$ and $w_B$ is significantly reduced. However, this method does **not** decrease the computation for calculating the gradient w.r.t. $x$. This is because it still needs to propagate the gradient through the weights $w$ to $x$, which considerably limits the performance of LoRA-based methods. As demonstrated in Figure 3, our LBP-WHT, requires much less computation while having better accuracy than LoRA-based methods; this is because our method reduces the computation for all procedures in BP, including the gradient calculation for both input and weights.

**Other Orthogonal Methods for On-device Training:** Previous research on efficient on-device model adaptation falls into two main categories. The first category [33, 38–43] suggests reducing the computational cost of arithmetic operations (addition and multiplication) in BP through quantization. The second category [20, 44] proposes to append a smaller neural network to the original model and accelerate adaptation by only training the attachment. To the best of our knowledge, our paper is the first to apply low-rank BP for ViT model adaptation. Therefore, our method, LBP-WHT, is distinct from previous research and can be combined with those methods for enhanced performance.

## 6  Conclusion

In this paper, we have addressed the problem of efficient model adaptation for ViT. We have proposed a novel low-rank BP technique designed to reduce the computational load associated with the propagation of gradients through the linear layers of ViT, which is a significant bottleneck when fine-tuning ViT. In Section 3, we introduced the LBP-WHT method as a solution to accelerate model adaptation. More specifically, LBP-WHT operates by projecting the gradient w.r.t. the output ($g_y$) into a low-rank space, performing matrix multiplications within this low-rank space, and then projecting the results back into the original space. Since all matrix multiplications occur in a low-rank space, the computational cost is significantly reduced. Additionally, thanks to the properties of the Walsh-Hadamard Transform (WHT), the overhead for these projections is minimal (as discussed in Section 3.3). Through extensive experiments in Section 4, we have demonstrated the efficiency and broad applicability of our method. Our LBP-WHT approach consistently outperforms existing methods with a significant speedup and higher accuracy.

## Acknowledgments and Disclosure of Funding

This work was supported in part by the US National Science Foundation (NSF) grant CNS-2007284.

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
