# Appendix for Efficient Low-rank Backpropagation for Vision Transformer Adaptation

## A    More Experimental Results for "Full Training" in Table 2 (Section 4.2)

Table 5 shows more results for training the entire model. For all models, our LBP-WHT consistently achieves both higher accuracy and lower computational cost (marked with ★★ in Table 5) than the baseline. Indeed, these results further demonstrate the effectiveness of our LBP-WHT approach.

| | | | | | | **Full Training** | | | | | |
|---|---|---|---|---|---|---|---|---|---|---|---|
| **Model** | **Method** | **R** | **Speedup** | **mAcc** | **MFLOPs** | **CF100** | **CF10** | **Cars** | **Flowers** | **Food** | **Pets** |
| Efficient Former L1 (Hybrid) | Full BP | - | 1.0 | 90.61 | 5841.09 | 84.72 | 96.88 | 87.84 | 95.48 | 85.70 | 93.05 |
| | LoRA-all | 8 | 1.5 | 89.13 | 4019.08 | 83.30 | 96.89 | 83.91 | 93.58 | 84.15 | 92.97 |
| | $LP_{L_1}$-4 | 10 | **2.7** | 84.30 | 2150.55 | 77.51 | 94.17 | 69.58 | 93.72 | 78.53 | 92.31 |
| | $LP_{L_1}$-6★★ | 21 | **1.7** | **89.55** | 3371.43 | 83.07 | 96.39 | 85.74 | 95.10 | 84.06 | 92.94 |
| | $LP_{L_1}$-7 | 28 | 1.4 | **89.96** | 4147.60 | 83.55 | 96.68 | 86.52 | 94.86 | 84.76 | 93.38 |
| | $LP_{L_1}$-8 | 36 | 1.2 | **90.03** | 5036.63 | 83.78 | 96.81 | 86.42 | 94.83 | 84.97 | 93.38 |
| Efficient Former L7 (Hybrid) | Full BP | - | 1.0 | 93.20 | 43128.48 | 88.54 | 98.20 | 91.10 | 97.64 | 89.36 | 94.36 |
| | LoRA-all | 8 | 1.6 | 92.08 | 26222.33 | 88.13 | 98.12 | 88.09 | 96.65 | 87.82 | 93.68 |
| | $LP_{L_1}$-4 | 10 | **3.4** | 91.69 | 12656.41 | 86.19 | 97.51 | 88.30 | 97.19 | 86.67 | 94.25 |
| | $LP_{L_1}$-6★★ | 21 | **1.9** | **92.54** | 22172.82 | 87.63 | 97.96 | 89.74 | 97.50 | 87.81 | 94.58 |
| | $LP_{L_1}$-8 | 36 | 1.2 | **92.79** | 35147.13 | 87.76 | 98.04 | 90.49 | 97.53 | 88.50 | 94.41 |
| Efficient FormerV2 S0 (Hybrid) | Full BP | - | 1.0 | 89.19 | 2259.93 | 84.06 | 96.88 | 84.80 | 93.62 | 84.99 | 90.79 |
| | LoRA-all | 8 | 1.2 | 86.07 | 1899.99 | 81.14 | 96.27 | 76.25 | 90.60 | 81.88 | 90.27 |
| | $LP_{L_1}$-4 | 10 | **1.9** | 78.56 | 1186.67 | 72.93 | 92.67 | 51.14 | 90.68 | 74.62 | 89.34 |
| | $LP_{L_1}$-6★★ | 21 | **1.4** | **86.52** | 1577.43 | 81.66 | 96.16 | 76.74 | 91.48 | 82.74 | 90.32 |
| | $LP_{L_1}$-7★★ | 28 | **1.2** | **87.86** | 1833.31 | 83.14 | 96.53 | 80.69 | 92.21 | 83.76 | 90.84 |
| | $LP_{L_1}$-8 | 36 | 1.1 | **88.56** | 2116.41 | 83.42 | 96.76 | 83.00 | 92.75 | 84.27 | 91.14 |
| Efficient FormerV2 L (Hybrid) | Full BP | - | 1.0 | 93.40 | 12614.40 | 89.37 | 98.56 | 91.18 | 96.81 | 89.49 | 94.96 |
| | LoRA-all | 8 | 1.4 | 92.37 | 8896.07 | 88.99 | 98.44 | 88.11 | 95.53 | 88.41 | 94.74 |
| | $LP_{L_1}$-4 | 10 | **2.5** | 87.51 | 4981.08 | 82.73 | 96.02 | 73.59 | 95.63 | 82.35 | 94.74 |
| | $LP_{L_1}$-6★★ | 21 | **1.7** | **92.40** | 7575.79 | 88.09 | 98.20 | 88.96 | 96.11 | 87.93 | 95.12 |
| | $LP_{L_1}$-8 | 36 | 1.1 | **93.18** | 11114.21 | 89.23 | 98.41 | 90.85 | 97.06 | 88.67 | 94.85 |
| SwinV2 Small (ViT) | Full BP | - | 1.0 | 93.77 | 48318.40 | 89.22 | 98.51 | 92.26 | 98.02 | 89.71 | 94.90 |
| | LoRA | 8 | 1.8 | 92.44 | 27202.90 | 87.62 | 98.15 | 87.81 | 96.24 | 90.24 | 94.60 |
| | LoRA-all | 8 | 1.7 | 92.78 | 27929.60 | 87.79 | 98.28 | 88.75 | 96.41 | 90.68 | 94.77 |
| | $LP_{L_1}$-4 | 10 | **2.5** | 91.07 | 19341.06 | 84.50 | 96.31 | 89.11 | 97.93 | 83.85 | 94.69 |
| | $LP_{L_1}$-6★★ | 21 | **1.9** | **93.37** | 25894.42 | 89.17 | 98.36 | 90.55 | 98.02 | 89.32 | 94.82 |
| | $LP_{L_1}$-8 | 36 | 1.4 | **93.88** | 34860.07 | 89.20 | 98.41 | 91.85 | 98.39 | 90.62 | 94.82 |

Table 5: Additional results for "Full Training" in Table 2. "$LP_{L_1}$-$r$" refers to our LBP-WHT method with $LP_{L_1}$-$r$ base selection as outlined in Equation 8. "mAcc" represents the mean accuracy across all datasets. "R" is short for "rank". "Hybrid" represents CNN-ViT-hybrid architecture. Results outperforming both LoRA and LoRA-all in speed and mAcc are underlined and marked with ★. Those exceeding all LoRA methods get ★★. Any results that have higher speed or mAcc are highlighted in bold.

## B    Compatibility with other orthogonal efficient training techniques (Section 5)

To support our claim that our method is complementary to other existing methods, we combine our LBP-WHT with LoRA and present our experimental results for training the last stage (partial training) of EfficientFormer-L1 in Table 6.

| **Method** | **GFLOPs** | **Memory [MB]** | | **Accuracy [%]** | |
|---|---|---|---|---|---|
| | | **Activation** | **Gradient** | **CF100** | **CF10** |
| Full BP | 121 | 141 | 2352 | 79.28 | 95.23 |
| LoRA-all | 62 | 142 | 44 | 76.92 | 94.38 |
| $LP_{L_1}$-2+LoRA-all | 4 | 9 | 44 | 73.27 | 92.62 |
| $LP_{L_1}$-4+LoRA-all | 13 | 29 | 44 | 75.48 | 93.74 |
| $LP_{L_1}$-8+LoRA-all | 48 | 104 | 44 | 76.58 | 94.33 |

Table 6: Results for combining our LBP-WHT with LoRA method on EfficientFormer-L1. "$LP_{L_1}$-$r$" refers to our LBP-WHT method with $LP_{L_1}$-$r$ base selection as outlined in Equation 8.

As shown in Table 6, our method significantly reduces both the storage size needed for the activation map ($x$ in Equation 1) and the computational costs. On the other hand, LoRA efficiently reduces the memory usage needed to store the weights gradient. By combining both methods, we can systematically reduce both computation and memory costs, while maintaining the accuracy levels close to using LoRA alone. For instance, when combining LBP-WHT with $LP_{L_1}$-4 base selection and LoRA, we achieve a speedup of 4.7x and memory savings of 2.5x, with only a slight accuracy drop of 1.4% compared to using LoRA alone. These results confirm the effectiveness of our method.

## C Evaluation on large scale dataset Places365

We test our method on a large-scale dataset Places365 [45], which contains over 1.8M training images and is more challenging than ImageNet (i.e., models have a lower accuracy on Places365 than ImageNet).

| Method | Speedup | MFLOPs | Accuracy [%] |
|---|---|---|---|
| Full BP | 1.0× | 1685.01 | 55.30 |
| LoRA | 6.9× | 242.61 | 50.64 |
| LoRA-all | 1.7× | 976.50 | 53.73 |
| $LP_{L_1}$-2 | 7.2× | 233.62 | 52.87 |
| $LP_{L_1}$-4 | 3.5× | 480.00 | 55.07 |
| $LP_{L_1}$-6 | 2.1× | 820.11 | 55.13 |
| $LP_{L_1}$-8 | 1.2× | 1397.02 | 55.39 |

Table 7: Evaluation results for partial training (training the last stage) of EfficientFormer-L1 on Places365 dataset. "$LP_{L_1}$-$r$" refers to our LBP-WHT method with $LP_{L_1}$-$r$ base selection as outlined in Equation 8.

As shown in Table 7, our method scales well on large scale datasets. For example, LBP-WHT with $LP_{L_1}$-2 base selection outperforms LoRA in both speed and accuracy; $LP_{L_1}$-8 has an even higher accuracy than the full-rank BP while achieving a $1.2\times$ speedup.

## D Preliminary Latency Evaluation on Edge Devices (Section 4)

| | | | EfficientFormer-L1 | | | | | | | EfficientFormer-L7 | | | |
|---|---|---|---|---|---|---|---|---|---|---|---|---|---|
| $(C_x, C_y, L)$ | Method | R | Speedup | | Latency [$\mu$s] | | $(C_x, C_y, L)$ | Method | R | Speedup | | Latency [$\mu$s] | |
| | | | CPU | GPU | CPU | GPU | | | | CPU | GPU | CPU | GPU |
| (448,1792,49) | Full BP | - | - | - | 8622.28 | 1.34 | (768,3072,49) | Full BP | - | - | - | 23390.21 | 3.49 |
| | $LP_{L_1}$-2 | 3 | 2.2× | 1.8× | 3862.15 | 0.73 | | $LP_{L_1}$-2 | 3 | 1.5× | 2.1× | 15835.63 | 1.65 |
| | $LP_{L_1}$-4 | 10 | 1.5× | 1.5× | 5681.61 | 0.88 | | $LP_{L_1}$-4 | 10 | 1.5× | 1.7× | 15376.71 | 2.04 |
| | $LP_{L_1}$-6 | 21 | 1.6× | 1.4× | 5539.20 | 0.96 | | $LP_{L_1}$-6 | 21 | 1.4× | 1.5× | 16754.33 | 2.28 |
| (1792,448,49) | Full BP | - | - | - | 8068.24 | 1.35 | (3072,768,49) | Full BP | - | - | - | 22193.53 | 3.50 |
| | $LP_{L_1}$-2 | 3 | 1.4× | 1.6× | 5666.05 | 0.87 | | $LP_{L_1}$-2 | 3 | 1.5× | 1.9× | 14423.38 | 1.85 |
| | $LP_{L_1}$-4 | 10 | 1.4× | 1.3× | 5750.53 | 1.03 | | $LP_{L_1}$-4 | 10 | 1.6× | 1.6× | 14108.66 | 2.23 |
| | $LP_{L_1}$-6 | 21 | 1.2× | 1.2× | 6858.44 | 1.12 | | $LP_{L_1}$-6 | 21 | 1.3× | 1.4× | 16950.27 | 2.45 |

Table 8: Latency for BP through the last two linear layers in EfficientFormer-L1 and L7. We implement our method with OpenBLAS and CuBLAS for deployment on CPU and GPU of NVIDIA Jetson Nano, respectively.

Table 8 shows the latency results for BP through the last two linear layers in EfficientFormer-L1 and L7 measured on NVIDIA Jetson Nano. Of note, our main contribution is on the algorithmic side and results in Table 8 are shown only for proving the potential of our approach for real deployment. We note that despite our naive implementation, our method still significantly out-performs the highly-optimized baseline methods.