# OpenReview forum: "Efficient Low-rank Backpropagation for Vision Transformer Adaptation"
_NeurIPS.cc/2023/Conference — NeurIPS 2023 poster_

### Official Review · Reviewer_Hdqi · 2023-06-12

**Soundness:** 3 good
**Presentation:** 3 good
**Contribution:** 2 fair
**Rating:** 4
**Confidence:** 4

**Summary:**

This paper proposed an efficient training method, called Low-rank Back Propagation via Walsh-Hadamard Transformation (LBP-WHT). Specifically, unlike previous works that target low-rank adaptation for model parameters, LBP-WHT aims to reduce the computational cost during backpropagation by projecting the gradients w.r.t inputs into a low-rank space, thus the overall computation can be done efficiently. Extensive experiments on classification and semantic segmentation have demonstrated that LBP-WHT significantly reduces MFLOPs during partially training.

**Strengths:**

1. To my knowledge, adopting low-rank backpropagation for ViT adaptation is novel.
2. The method is easy to follow. The visualization of LBP-WHT in Figure 2 clearly presents the proposed framework.
3. The results are impressive. Based on partially training (Table 2), LBP-WHT achieves great efficiency-accuracy trade-off.

**Weaknesses:**

1. Theoretical complexity reduction cannot reflect the real speedup for training. It would be better for the authors to report training time (GPU hours) or training throughput (images/s) in main experiments, especially for Table 2. Besides, as shown in Table 5 of the supplementary material, the authors have compared latency across different hardwares. However, the real speedup is only around 1-2x, which is very different from the reduction ratio of MFLOPs (6-7x).
2. The experiments do not report memory cost. Recent works (especially for LLMs) usually care more about memory cost during adaptation since fully finetuning or low-rank finetuning requires intensive memory. Moreover, on-device training also requires low-memory consumption, see [33]. However , the effect of the propsoed method on memory cost is also unknown.
3. As this paper aims to reduce the computational cost during backpropagation, it does not have advantages in inference. Actually, it even introduces overhead on model parameters as it is not parameter-efficient on downstream tasks.
4. Lack ablation study of combining LoRA with LBP-WHT. Since authors have claimed that the proposed technique is complementary to other works, it should have at least one experiment to prove this claim and show the advantage of their method.

**Questions:**

1. In Table 3, why LoRA and LBP-WHT have much more MFLOPs than Full BP on partial training?
2. Can authors also explain why the proposed method performs better on partial training?

**Limitations:**

Yes

---

> ### Author Rebuttal · Authors · 2023-08-09
>
> 1. > *Theoretical complexity reduction cannot reflect the real speedup for training. It would be better for the authors to report training time (GPU hours) or training throughput (images/s) in main experiments, especially for Table 2.*
>
>    As our primary contribution is on the algorithm side, we use #FLOPs as a measure of performance which is **independent of the implementation details**. This approach enables us to determine the acceleration resulting from a more efficient algorithm, not simply from an improved implementation.
>
>    $~$
>
>    As noted in the supplementary material, our current implementation is only meant to be a **proof of concept**, hence it is not fully optimized. As such, we are aware of several limitations:
>
>    - The WHT is carried out using general matrix multiplications. Consequently, none of the properties of WHT are exploited, including the fast WHT algorithm.
>    - There is a high degree of “algorithm-oblivious” overhead. While the baseline method has been integrated into a single CUDA kernel, our method launches multiple CUDA kernels, thus leading to considerable context switching and kernel launching overhead [50].
>
>    $~$
>
>    We found that the above limitations do **not** scale with the problem size. Therefore, to demonstrate the effectiveness of our method, we test it with the linear layers from larger scale models. The results, measured on a RTX3090Ti, are as follows:
>
>    |Token Dimension|Projection Rank|Theoretical Speedup|Actual Speedup|Baseline Run Time [ms]|LBP-WHT Run time [ms]|
>    |-|-|-|-|-|-|
>    |2048|16|3.94x|3.26x|0.66|0.20|
>    |2048|32|1.97x|1.84x|0.66|0.36|
>    |4096|16|3.97x|3.90x|3.18|0.82|
>    |4096|32|1.98x|1.96x|3.18|1.63|
>
>    As shown, the speedup achieved by our method closely aligns with the theoretical speedup; this confirms that our method is practical and can be efficiently implemented.
>
>    ---
>
> 2. > *Does LBP-WHT have benefit in memory savings? Can LBP-WHT be combined with other efficient training methods?*
>
>    Our method offers benefits in memory usage. As shown in Algorithm 1, the activation map 'x' is projected into a low-rank space through the use of WHT. This step notably reduces the memory needed to store the activation map.
>
>    $~$
>
>    Finally, our method can be paired with other techniques for additional memory savings, *e.g.*, we can combine our approach with LoRA to reduce memory consumption for both activations and gradients - table below. Meaning of notations is introduced in the general rebuttal section [*[link]*](https://openreview.net/forum?id=NNtsO5L27J&noteId=8YCHi4VqDZ).
>
>    |Method|Accuracy CF100 [%]|Accuracy CF10 [%]|Activation [MB]|Gradient [MB]|GFLOPs|
>    |-|-|-|-|-|-|
>    |Full BP|79.28|95.23|141|2352|121|
>    |LoRA-all|76.92|94.38|142|44|62|
>    |$LP_{L_1}-2$|75.61|93.35|9|2352|8|
>    |$LP_{L_1}-4$|78.27|94.60|29|2352|25|
>    |$LP_{L_1}-8$|79.34|95.31|104|2352|91|
>    |$LP_{L_1}-2$+LoRA-all|73.27|92.62|9|44|4|
>    |$LP_{L_1}-4$+LoRA-all|75.48|93.74|29|44|13|
>    |$LP_{L_1}-8$+LoRA-all|76.58|94.33|104|44|48|
>
>    As shown, our method significantly reduces both the storage needed for the activation map and the computational costs. By combining with LoRA, our method can **systematically reduce** both computation and memory costs, while maintaining accuracy close to using LoRA alone. For instance, when combining LBP-WHT $LP_{L_1}$-4 and LoRA, we achieve a speedup of 4.7x and memory savings of 2.5x, with only a slight accuracy drop of 1.4% compared to employing LoRA alone. These results confirm the effectiveness of our method.
>
>    ---
>
> 3. > *As this paper aims to reduce the computational cost during backpropagation, it does not have advantages in inference.*
>
>    Our method is designed to improve efficiency during *back-propagation* rather than during inference. This is because, during the training phase, back propagation requires twice as much computation as inference, thus it is the main bottleneck.
>
>    $~$
>
>    Interestingly enough, our focus on backpropagation offers another **advantage**. Modifying the operators involved in inference typically demands re-training the parameters in the modified operator from scratch, which brings about a substantial overhead and minimizes the benefits of efficient model adaptation. In contrast, our method only modifies the back propagation process while leaving the inference process untouched. Hence our method can be directly applied to any pre-trained model, thus enabling immediate model adaptation.
>
>    ---
>
> 4. > *Actually, it even introduces overhead on model parameters as it is not parameter-efficient on downstream tasks.*
>
>    We want to clarify that:
>
>    - Our method introduces **no extra** trainable parameter to the model. WHT base matrix is fixed regardless of the model and dataset.
>    - Although the WHT base matrix might appear as a "parameter", it is primarily used for theoretical analysis. For ease of understanding, we cast WHT as the process of multiplying the gradient map with this base matrix. However, in practice, the fast-WHT algorithm is employed. This algorithm integrates the base matrix into a butterfly diagram, thus eliminating the need to store the base matrix separately. This is analogous to the Discrete Fourier Transformation (DFT). While DFT can be theoretically expressed as matrix multiplication, in practical applications, the Fast Fourier Transformation (FFT) is preferred.
>
>    ---
>
> 5. > *In Table 3, why LoRA and LBP-WHT have much more MFLOPs than Full BP on partial training?*
>
>    Thank you for pointing this out. We made a mistake when filling this table. The correct computation required by Full BP is 10052.00 MFLOPs. We’ll fix it in the camera ready version.
>
>    ---
>
> 6. > *Can authors also explain why the proposed method performs better on partial training?*
>
>    In model adaptation, freezing some layers is important to solve the overfitting problem [51, 52]. Our experimental results align with the general model adaptation problems.

---

### Official Review · Reviewer_EswK · 2023-07-05

**Soundness:** 3 good
**Presentation:** 3 good
**Contribution:** 3 good
**Rating:** 5
**Confidence:** 5

**Summary:**

This paper introduces a method to efficiently finetune Vision Transformers (ViTs) for specific tasks such as image classification and semantic segmentation. The method, Low-rank Backpropagation via Walsh-Hadamard Transformation (LBP-WHT), reduces the computational cost of backpropagation in ViTs by projecting the gradients (dy/dx and dy/dw) into a low-rank space using WHT basis where the bases are made up of -1's and 1's. Experiments are only shown with EfficientFormer (V1, V2), SwinTransformerV2 and SegFormer models. Proposed method outperforms LoRA in terms of both accuracy and efficiency. The paper also shows actual speed improvements on hardware (GPU/CPU) which is promising.

**Strengths:**

### Simple method with significant speedups and possible efficient implementation
* The proposed method is simple in design and easy to implement *efficiently* on any target hardware. The authors have already shown speed-ups on CPU and GPU with a naive implementation which I believe can be sped up for commercial deployment

### Good results compared to LoRA
* The experiments demonstrate strong results on EfficientFormer (V1,V2), SwinTransformer and SegFormer models as compared to LoRA. The proposed method achieves better speed-up than LoRA while also not losing as much accuracy.

### Can be applied to both dy/dw and dy/dx
* LoRA cannot speed-up the gradient computation for dy/dx but the proposed method can LBP-WHT.

**Weaknesses:**

### Why only ViTs? That too only smaller scale ViT models?
* The proposed method is an efficient variant of backpropagation that should work with any architecture that contains linear layers. So, why not evaluate the method with MLP-Mixer models (Tolstikhin, Ilya O., et al. "Mlp-mixer: An all-mlp architecture for vision." NeurIPS 2021) which are full of linear layers?
* Also, many CNN models (such as all MobileNets, EfficientNets) use pointwise convolution layers and pointwise convolutions are basically linear layers. The authors should show results of the proposed method with such architectures too, not just ViTs.
* How does it compare with LoRA on Large Language Models? The LoRA paper showed exceptional results with LLMs, but in this paper, the proposed method has been compared with LoRA only on small-scale ViTs.

### Comparison with Gradient Compression
* The main contribution of this paper is a method to reduce the cost of computing gradients, which they do so by computing low-rank approximations. There is a popular method called Gradient Compression (Lin, Yujun, et al. "Deep Gradient Compression: Reducing the Communication Bandwidth for Distributed Training." ICLR 2018) which compresses gradients by 270x to 600x without losing accuracy. Comparison with such a method should be shown in the results.

**Questions:**

* Do you use your method in all linear layers or only the attention modules (like "LoRA baseline")? My guess is all linear layers, but I don't see this explicitly mentioned.

**Limitations:**

Yes, limitations have been sufficiently addressed.

---

> ### Author Rebuttal · Authors · 2023-08-09
>
> 1. > *Why only ViTs? That too only smaller scale ViT models?*
>
>    Yes, LBP-WHT works well with other architectures. In this paper, both EfficientFormer and EfficientFormerV2 are hybrid models of CNN-ViT that incorporate both multi-head attention modules from ViT and convolution layers from CNN. To elaborate, smaller models such as EfficientFormer-L1 and EfficientFormerV2-S0 consist of more convolutional layers than multi-head attention modules. On the other hand, larger models like EfficientFormer-L7 and EfficientFormerV2-L feature more multi-head attention modules. As shown in Table 2 of our paper, the LBP-WHT method works effectively with **all** these hybrid models. This confirms that our technique is efficient in handling both multi-head attention modules and convolutional layers.
>
>    ---
>
> 2. > *Comparison with Gradient Compression*
>
>    Deep Gradient Compression concentrates on enhancing communication efficiency in multi-node training scenarios and addresses the challenge of aggregating gradients from each device efficiently **after** back propagation. In contrast, our focus is on the efficient execution of the back propagation itself. As these approaches target different optimization goals and are applied at distinct stages of training, a direct comparison between our method and Deep Gradient Compression is not meaningful.
>
>    ---
>
> 3. > *Do you use your method in all linear layers or only the attention modules (like "LoRA baseline")? My guess is all linear layers, but I don't see this explicitly mentioned.*
>
>    Yes, our method is applied to **all** layers, including the linear layer for the self-attention module, as well as the linear layers in the subsequent feed-forward network. For the baseline method, LoRA, we conducted two sets of experiments labeled as "LoRA" and "LoRA-all". The "LoRA" experiments apply the LoRA method only to the *self-attention layers*, consistent with the experimental setup proposed in the original LoRA paper. In contrast, "LoRA-all" applies the LoRA method to all *linear* *layers*, including those in the self-attention modules and the feed-forward network.

---

> > ### Comment · Reviewer_EswK · 2023-08-13
> >
> > Currently, my understanding is that the proposed method is only applicable to Linear layers. So,
> > * is it correct to say that it cannot be applied to convolution layers?
> > * In that case, even if you apply it to a hybrid architecture like EfficientFormer, are you only applying this low-rank backpropagation to the linear layers in the architecture and NOT to any of the convolution layers?
> > * And if that is true, how do you backpropagate through the conv layers? Is it just vanilla backpropagation?
> > The paper currently lacks a detailed analysis and discussion of these aspects.
> >
> > ---
> >
> > Since LoRA is the only point of comparison, can you please answer my comment about comparisons on finetuning large language models?

---

> > > ### Author Response · Authors · 2023-08-16
> > > **Response to Reviewer EswK**
> > >
> > > 1. >*Currently, my understanding is that the proposed method is only applicable to Linear layers. So, is it correct to say that it cannot be applied to convolution layers? In that case, even if you apply it to a hybrid architecture like EfficientFormer, are you only applying this low-rank backpropagation to the linear layers in the architecture and NOT to any of the convolution layers? And if that is true, how do you backpropagate through the conv layers? Is it just vanilla backpropagation? The paper currently lacks a detailed analysis and discussion of these aspects.*
> > >
> > >     We want to emphasize that our technique can be used with *any layer* computed with Matrix Multiplication, including convolutional layers. That is, if an operation can be computed with matrix multiplication, then our method is applicable. Consequently, we have implemented and evaluated our method across all these types of layers, including linear and convolutional layers, in the EfficientFormer and EfficientFormerV2 models - see our results in Table 2 and Figure 3 in the paper.
> > >
> > >     $~$
> > >
> > >     To provide more intuition, let’s take convolution as an example: A convolutional layer is essentially a combination of a linear layer (matrix multiplication) and an im2col operation [53], which can be represented as $y = w * x = w \cdot \text{im2col}(x)$ where variables $x$, $y$, $w$ refer to input, output and weights, respectively. Meanwhile, the operators “$*$” and “$\cdot$” refer to convolution and matrix multiplication, respectively. In PyTorch,  the convolutional layer can be described as follows:
> > >
> > >     ```python
> > >     # Input x and weights w
> > >     x = torch.randn((batch_size, input_channel, height, width))
> > >     w = torch.randn((output_channel, input_channel, kernel_size, kernel_size))
> > >     # Perform Im2col and put the result in x_im2col
> > >     x_im2col = torch.nn.functional.unfold(x, kernel_size=kernel_size, padding=kernel_size//2)
> > >     # Perform matrix multiplication and save the result in y_im2col
> > >     y_im2col = torch.matmul(w.view(output_channel, -1), x_im2col)
> > >     # Recover the shape of y from y_im2col
> > >     y = y_im2col.view(batch_size, output_channel, height, width)
> > >     ```
> > >
> > >     Here, the `torch.nn.functional.unfold` serves as the im2col function in PyTorch and `torch.matmul` serves as matrix multiplication. This way, we can seamlessly apply our approach to any layer computed with matrix multiplication by replacing the back-propagation operation for the matrix multiplication with our LBP-WHT.
> > >
> > > ---
> > >
> > >
> > > 2. >*Since LoRA is the only point of comparison, can you please answer my comment about comparisons on finetuning large language models?*
> > >
> > >     We aim to broaden the scope of our research beyond just computer vision tasks, potentially exploring areas like language models in future. That said, our current findings, as highlighted in the experimental section, already demonstrate that our LBP-WHT is highly effective across a diverse range of models such as pure ViT and hybrid CNN-ViT, and for different tasks like image classification and segmentation. Given these outcomes, we are optimistic about the applicability of LBP-WHT to areas beyond computer vision.
> > >
> > >     $~$
> > >
> > >     Furthermore, the role of fine-tuning in vision models cannot be understated. Vision Transformers (ViTs) often find themselves in environments where the data distribution diverges significantly from their original training datasets. In such situations, it becomes paramount to fine-tune these models efficiently, enabling them to acclimate to the new context; this underscores the relevance of our research. Take, for instance, the use of ViTs in smartphones for scene understanding. On-device training of ViTs lets us harness user-specific data without compromising on privacy. Here, our LBP-WHT becomes instrumental as it facilitates viable training of ViTs on-device. This ensures that state-of-the-art computer vision models become accessible to a broader audience.

---

### Official Review · Reviewer_idrd · 2023-07-07

**Soundness:** 3 good
**Presentation:** 2 fair
**Contribution:** 3 good
**Rating:** 6
**Confidence:** 4

**Summary:**

The paper presents to compute the low-rank approximation for the matrices used in back-propagation, and thus it can save the computation in multiplication. The proposed approach is more efficient than LoRA because no need to backpropagate to the original weights, and experiments also support the claims.

**Strengths:**

1. The proposed approach is simple and well-motivated.
2. The experimental results show that the approach is better than LoRA.

**Weaknesses:**

1. Although the paper claims that the approach is complementary to other efficient approaches (such as quantization), there is no experimental support for it.

**Questions:**

1. In Table 1, what is the difference between the r in "LPL1 -r” and R? I thought they both represent rank.
2. Is the WHT differentiable? How is the model trained with this operation.
3. I thought WHT^{-1} was not a square matrix, was it? If not, how to compute WHT^{-1}?
4. How is the "Speed up" in table 1 measured?

---
**Post-rebuttal**
Thank you for the authors' response. I have read it and it addressed my questions.

**Limitations:**

1. There is no experiment to support whether the approach can work well with other efficient approaches.
2. The experimental results are only for vision tasks.

---

> ### Author Rebuttal · Authors · 2023-08-09
>
> 1. > *Although the paper claims that the approach is complementary to other efficient approaches (such as quantization), there is no experimental support for it.*
>
>    **Experimental results when combining LBP-WHT with LoRA**
>
>    To show that our method is complementary to other existing methods, we combine LBP-WHT with LoRA and show our results below. Meaning of notations "LoRA", "LoRA-all" and $LP_{L_1}$-r is introduced in the general rebuttal section [*[link]*](https://openreview.net/forum?id=NNtsO5L27J&noteId=8YCHi4VqDZ).
>
>    |Method|Accuracy CF100 [%]|Accuracy CF10 [%]|Activation [MB]|Gradient [MB]|GFLOPs|
>    |-|-|-|-|-|-|
>    |Full BP|79.28|95.23|141|2352|121|
>    |LoRA-all|76.92|94.38|142|44|62|
>    |$LP_{L_1}-2$|75.61|93.35|9|2352|8|
>    |$LP_{L_1}-4$|78.27|94.60|29|2352|25|
>    |$LP_{L_1}-8$|79.34|95.31|104|2352|91|
>    |$LP_{L_1}-2$+LoRA-all|73.27|92.62|9|44|4|
>    |$LP_{L_1}-4$+LoRA-all|75.48|93.74|29|44|13|
>    |$LP_{L_1}-8$+LoRA-all|76.58|94.33|104|44|48|
>
>    As shown, our method significantly reduces both storage size needed for activation map (x) and computational costs. LoRA reduces the memory usage needed to store the gradients. By combining both methods, we can **systematically reduce** both computation and memory costs, while maintaining the accuracy close to using LoRA alone. For instance, when combining LBP-WHT with $LP_{L_1}$-4 base selection and LoRA, we achieve a speedup of 4.7x and memory savings of 2.5x, with only an accuracy drop of 1.4% compared to using LoRA alone. These results confirm the complementarity and effectiveness of our method.
>
> ---
> 2. > *In Table 1, what is the difference between the r in "LPL1 -r” and R? I thought they both represent rank.*
>
>    The "r" in $LP_{L_1}$-r is the hyper-parameter associated with $L_1$-based low-pass basis selection in Equation (8). As Figure 2(c) shows, each WHT basis is linked with a 2D coordinate (i, j). A larger i or j value correlates with a higher frequency change in one direction. The $LP_{L_1}$ selection method picks WHT bases with a smaller $L_1$ norm of the coordinate. For instance, $LP_{L_1}$-2 selects bases with the $L_1$ norm of the indices less than 2, that is, $|i| + |j| < 2$, such as (0, 0), (0, 1), and (1, 0).
>
>    $~$
>
>    On the other hand, "R" in Table 1 refers to the rank of the low-rank projection. For example, since $LP_{L_1}$-2 selects three WHT bases, the rank for the low-rank projection $LP_{L_1}$-2 is therefore three and R=3.
>
>    We will clarify these notations in the camera-ready version.
>
> ---
> 3. > *Is the WHT differentiable? How is the model trained with this operation.*
>
>    While the Walsh-Hadamard Transformation (WHT) is differentiable, this property is **not required** for our method, *i.e.*, our method does **not** affect the forward propagation process. Instead, our approach modifies and accelerates the *back propagation*. With our method, the training process goes as follows:
>    - Given the inputs and weights, perform forward propagation as usual
>    - Calculate the loss value as usual
>    - Replace the back propagation (or auto-differentiation) with LBP-WHT and calculate the gradient with our method.
>
>    Since our method only modifies back propagation, LBP-WHT can be applied in a plug-and-play manner for fine-tuning various models. Also, the LBP-WHT does **not** affect the inference performance.
>
> ---
> 4. > *I thought WHT^{-1} was not a square matrix, was it? If not, how to compute WHT^{-1}?*
>
>    As shown in Equation (2), $WHT^{-1}$ refers to the reverse WHT **function**, *not* the inverse matrix. Similar to reverse Fourier Transformation, the reverse WHT is calculated by multiplying the low-rank gradient ($\hat g_x$) with a transposed WHT base.
>
> ---
> 5. > *How is the "Speed up" in table 1 measured?*
>
>    The speedup in Table 1 is determined by comparing the #FLOPs required by the baseline method to #FLOPs of our LBP-WHT method. As our primary contribution is on the algorithm side, we use #FLOPs as a measure of performance that is **independent of the implementation**. This enables us to determine the acceleration resulting from proposing a more efficient algorithm, not simply from using an improved implementation.
>
>    $~$
>
>    To demonstrate that the theoretical speedup translates into actual speedup, we profile the linear layers in a feed-forward network with different token dimensions on a RTX3090Ti GPU:
>
>    |Token Dimension|Projection Rank|Theoretical Speedup|Actual Speedup|Baseline Run Time [ms]|LBP-WHT Run time [ms]|
>    |-|-|-|-|-|-|
>    |2048|16|3.94x|3.26x|0.66|0.20|
>    |2048|32|1.97x|1.84x|0.66|0.36|
>    |4096|16|3.97x|3.90x|3.18|0.82|
>    |4096|32|1.98x|1.96x|3.18|1.63|
>
>    As shown, the actual speedup closely mirrors the theoretical speedup; this validates the efficiency of our method.
>
> ---
> 6. > *The experimental results are only for vision tasks.*
>
>    We plan to extend our work beyond computer vision applications, *e.g.*, language modeling, in our future research. However, our current results in the experimental section show already that our LBP-WHT works very efficiently with a wide class of models, (*e.g.*, pure ViT and hybrid CNN-ViT) and applications (*e.g.*, image classification and segmentation). Hence, we are confident that our LBP-WHT will generalize seamlessly to applications beyond computer vision.
>
>    $~$
>
>    We also want to emphasize the importance of fine-tuning vision models. ViTs have been deployed in scenarios where data distributions are very different from the datasets used to pretrain the model. In such cases, efficiently finetuning the model and helping the model adapt to the new domain is both **important and necessary**, hence the significance of our work. For instance, ViTs are utilized on smartphones for tasks such as scene comprehension. Training a ViT on a smartphone allows us to utilize the user data while maintaining privacy. In these scenarios, our LBP-WHT is crucial in making ViT training feasible, thus allowing everyone benefit from SoTA computer vision models.

---

### Official Review · Reviewer_j7ix · 2023-07-09

**Soundness:** 4 excellent
**Presentation:** 2 fair
**Contribution:** 3 good
**Rating:** 6
**Confidence:** 4

**Summary:**

The author discusses the issue of computational complexity during matrix multiplication in the backpropagation of linear layers in the vision transformer architecture. To address this issue, the author proposes a method called Low-rank BackPropagation via Walsh-Hadamard Transformation (LBP-WHT), which projects the gradient into a low-rank space and performs BackPropagation computations. Finally, the author evaluates the proposed method using different models such as ViT and a hybrid convolution-ViT model on multiple datasets. The author claims that LBP-WHT achieves 10.4% higher accuracy than the state-of-the-art baseline, providing SOTA performance on the CIFAR100 dataset with the EfficientFormer-L1 model.

**Strengths:**


1. This paper is well-written and easy to understand.

2. The Low-rank BackPropagation via Walsh-Hadamard Transformation (LBP-WHT) method in the gradient domain is new and effective.

3. The novelty of LBP-WHT as the first work to accelerate ViT training using low-rank BP (Backpropagation). It also demonstrates the ability to lower the computational barrier and enable the adaptation of large ViT models on resource-constrained edge devices.

4. The paper addresses an essential problem in transformer architecture and proposes a novel solution that achieves promising results.

5. The author provides more quantitative results on various tasks, including classification and semantic segmentation. Additionally, the comparison with LoRA, one of the state-of-the-art (SOTA) models, adds value to the evaluation.


**Weaknesses:**

Weaknesses:

1. The literature survey section in this paper is a bit weak. The author should compare related/similar transformer models, particularly QLoRA, and similar work on "Hadamard product for low-rank bilinear pooling." A more thorough literature review is necessary in this regard.

2. It would be beneficial to add a section briefly discussing LoRA, QLoRA, and the proposed method, as well as comparing the results with the proposed method.

3. The author should provide results on larger datasets like the standard ImageNet-1k and evaluate other large datasets such as JFT-Imagenet-21K.

4. Instead of Swin-Transformer, the author should evaluate the proposed method on the SOTA model like WaveViT or SpectFormer to validate the SOTA performance.

5. The paper misses one of the core aspects of machine learning practice: readability and reproducibility of results. It would be more transparent and beneficial if the author could provide source code to reproduce the results, similar to what was done with LoRA.

6. It seems like a minor typo: "finally project them black" should be "finally project them back."

**Questions:**

Please refer to the weakness section.

**Limitations:**

Please refer to the weakness section.

---

> ### Author Rebuttal · Authors · 2023-08-09
>
> 1. > *The literature survey section in this paper is a bit weak. The author should compare related/similar transformer models, particularly QLoRA, and similar work on "Hadamard product for low-rank bilinear pooling." A more thorough literature review is necessary in this regard. It would be beneficial to add a section briefly discussing LoRA, QLoRA, and the proposed method, as well as comparing the results with the proposed method.*
>
>    We appreciate your suggestions and will certainly expand the related work section, including a more detailed presentation of the LoRA method.
>
>    $~$
>
>    As for QLoRA [46], we note that QLoRA was released on arxiv only on May 23, 2023, which is *after* the NeurIPS 2023 paper submission deadline. Therefore, we were not aware of it at the time of our paper submission. Given our resources, we can’t provide such a comparison within the rebuttal period.
>
>    $~$
>
>    Regarding [47], despite the title similarity, we note that the model and the method used are entirely different than ours; more precisely:
>
>    - The method in [47] can only work with a bilinear model. In contrast, our method is designed for ViTs. Thus, a direct comparison between our method and method in [47] is simply not possible.
>    - The operator “Hadamard product” used in [47], which refers to element-wise multiplication, is fundamentally different from the Walsh-Hadamard Transformation (WHT) utilized in our paper. While the Hadamard product is a method for performing multiplication, the WHT is an orthogonal transformation method.
>
>    Given these fundamental differences, we cannot directly compare our method with methods used in [47].
>
>    ---
>
>
>
> 2. > *The author should provide results on larger datasets like the standard ImageNet-1k and evaluate other large datasets such as JFT-Imagenet-21K.*
>
>    **Experimental results on large-scale dataset**
>
>    We test our method on large-scale dataset Places365 [45], which contains over 1.8M training images with 365 different classes and is a more challenging dataset than ImageNet-1K (the same networks usually have a lower accuracy on Places365 than ImageNet-1K). We present below our experimental results for finetuning the EfficientFormer-l1 with partial training experimental setup (Section 4.1). Meaning of notations "LoRA", "LoRA-all" and $LP_{L_1}$-r is introduced in the general rebuttal section above [*[link to the notations table]*](https://openreview.net/forum?id=NNtsO5L27J&noteId=8YCHi4VqDZ).
>
>    |Method|R|Speedup|MFLOPs|Accuracy [%]|
>    |-|-|-|-|-|
>    |Full BP|-|1.0x|1685.01|55.30|
>    |LoRA|8|6.9x|242.61|50.64|
>    |LoRA-all|8|1.7x|976.50|53.73|
>    |$LP_{L_1}$-2|3|7.2x|233.62|52.87|
>    |$LP_{L_1}$-4|10|3.5x|480.00|55.07|
>    |$LP_{L_1}$-6|21|2.1x|820.11|55.13|
>    |$LP_{L_1}$-8|36|1.2x|1397.02|55.39|
>
>    As shown in this table, our method **scales well** for such large scale datasets. For example, LBP-WHT with $LP_{L_1}$-2 base selection outperforms LoRA both in speed and accuracy. As another example, $LP_{L_1}$-8 has even a higher accuracy than full-rank BP while achieving a 1.2x speedup.
>
>    $~$
>
>    **Experimental result for combining LBP-WHT with LoRA**
>
>    To support our claim that our method is complementary to other existing methods, we combine our LBP-WHT with LoRA and present below our experimental results for accuracy, computation and memory cost. Meaning of notations "LoRA", "LoRA-all" and $LP_{L_1}$-r is introduced in the general rebuttal section above [*[link to the notations table]*](https://openreview.net/forum?id=NNtsO5L27J&noteId=8YCHi4VqDZ).
>
>    |Method|Accuracy CF100 [%]|Accuracy CF10 [%]|Activation [MB]|Gradient [MB]|GFLOPs|
>    |-|-|-|-|-|-|
>    |Full BP|79.28|95.23|141|2352|121|
>    |LoRA-all|76.92|94.38|142|44|62|
>    |$LP_{L_1}-2$|75.61| 93.35| 9| 2352|8|
>    |$LP_{L_1}-4$|78.27| 94.60| 29| 2352|25|
>    |$LP_{L_1}-8$|79.34| 95.31| 104| 2352|91|
>    |$LP_{L_1}-2$ + LoRA-all|73.27|92.62|9|44|4|
>    |$LP_{L_1}-4$ + LoRA-all|75.48|93.74|29|44|13|
>    |$LP_{L_1}-8$ + LoRA-all|76.58|94.33|104|44|48|
>
>
>    As shown, our method significantly reduces both the storage size needed for the activation map (x) and the computational costs. On the other hand, LoRA efficiently reduces the memory usage needed to store the weights gradient. By combining both methods, we can **systematically reduce** both computation and memory costs, while maintaining the accuracy levels close to using LoRA alone. For instance, when combining LBP-WHT with $LP_{L_1}$-4 base selection and LoRA, we achieve a speedup of 4.7x and memory savings of 2.5x, with only a slight accuracy drop of 1.4% compared to using LoRA alone. These results confirm the effectiveness of our method.
>
>    ---
>
>
>
> 3. > *Instead of Swin-Transformer, the author should evaluate the proposed method on the SOTA model like WaveViT or SpectFormer to validate the SOTA performance.*
>
>    EfficientFormer [48] and EfficientFormer-V2 [49] were proposed in Oct 2022 and Dec 2022, respectively, and both of them were considered SoTA at the time of our paper submission.
>
>    $~$
>
>    Based on our experimental section, our LBP-WHT method works efficiently with a large range of models, such as pure ViT and hybrid CNN-ViT, and for various applications like classification and segmentation. Hence, we have confidence in generalizing our LBP-WHT approach seamlessly to other models and other applications.
>
>    ---
>
>
>
> 4. > *The paper misses one of the core aspects of machine learning practice: readability and reproducibility of results. It would be more transparent and beneficial if the author could provide source code to reproduce the results, similar to what was done with LoRA.*
>
>    Indeed, all experimental results reported in our paper are fully reproducible. We will open source the code upon acceptance.

---

> > ### Comment · Reviewer_j7ix · 2023-08-21
> >
> > After reviewing the rebuttal and considering the authors’ response to my concern about LBP-WHT with LoRA, I appreciate that the authors have addressed my concern. I acknowledge that the performance does improve using this experiment, but note that the improvement seems to be minimal. Overall, after considering the rebuttal and other reviews, you have decided to maintain your current rating.

---

> > > ### Author Response · Authors · 2023-08-21
> > > **Response to Reviewer j7ix**
> > >
> > > Thank you for your feedback. To clarify the benefits of integrating our method with LoRA, we present a comparison of computation and memory usage between using LoRA alone and using the combined approach of LoRA + LBP-WHT (our method).
> > >
> > > | Method                  | Memory Saving   | Speedup          | Memory [MB] | Computation [GFLOPs] |
> > > | ----------------------- | --------------- | ---------------- | ----------- | -------------------- |
> > > | LoRA-all                | -               | -                | 186         | 62                   |
> > > | $LP_{L_1}-2$ + LoRA-all | **4.3$\times$** | **15.5**$\times$ | 43          | 4                    |
> > > | $LP_{L_1}-4$ + LoRA-all | **2.5**$\times$ | **4.8**$\times$  | 73          | 13                   |
> > > | $LP_{L_1}-8$ + LoRA-all | **1.3**$\times$ | **1.3**$\times$  | 148         | 48                   |
> > >
> > > The results show that when LoRA is combined with our LBP-WHT method, we achieve up to a 15.5$\times$ increase in speed and save memory by a factor of 4.3$\times$. This significant improvement underscores the effectiveness of our approach.

---

### Official Review · Reviewer_F5LH · 2023-07-26

**Soundness:** 3 good
**Presentation:** 3 good
**Contribution:** 3 good
**Rating:** 5
**Confidence:** 4

**Summary:**

The paper proposes a cost-efficient back-propagation method using a low-rank trick for vision transformers. When transferring a huge-size vision transformer to the downstream task, the training cost would be high. The paper tackles the problem using low-rank approximation with Walsh-Hadamard Transformation (WHT). The experiments demonstrate the training efficiency of the proposed low-rank back-propagation.


**Strengths:**

- The paper presents their motivation and method well.
- Experiments show promising performance.
- Ablation studies are informative.

**Weaknesses:**

- The targeting scenario of the paper seems somewhat narrow. The paper focuses on vision transformers and its adaptation (transfer learning scenario). It would be nice to discuss the possibility of the proposed method for the efficient pre-training scenario. Also, a discussion about general transformer (e.g., language modeling transformer) adaptation beyond the vision domain would make the paper stronger.



**Questions:**

- The analysis of computation overhead is in section 3.3., but I could not find the cost of computing the low-rank projection matrix (p). It is neglectable, right?
- Could you provide more experiments on vanilla vision transformers?


**Limitations:**

The authors addressed limitation in the manuscript.

---

> ### Author Rebuttal · Authors · 2023-08-09
>
> 1. > *The targeting scenario of the paper seems somewhat narrow. The paper focuses on vision transformers and its adaptation. It would be nice to discuss the possibility of the proposed method for the efficient pre-training scenario. Also, a discussion about general transformer adaptation beyond the vision domain would make the paper stronger.*
>
>    We plan to extend our work beyond vision applications, *e.g.*, language models, in our future research. However, we note that our current results in the experimental section show already that the LBP-WHT method works very efficiently for a wide class of models, (*e.g.*, pure ViT and hybrid CNN-ViT) and applications (*e.g.*, image classification and segmentation). Hence, we are confident that our LBP-WHT will generalize seamlessly to applications beyond computer vision.
>
>    $~$
>
>    We also want to emphasize the importance of fine-tuning for vision models. ViTs have been deployed in scenarios where data distributions are very different from the datasets used during pretraining. In such cases, efficiently finetuning the model and helping the model adapt to the new domain is both **important and necessary**, thus proving the significance of our work. For instance, ViTs are utilized on smartphones for tasks such as scene comprehension. Training a ViT on a smartphone allows us to utilize the user data while maintaining privacy. In such scenarios, our LBP-WHT is crucial in making ViT training feasible, thus allowing more people to benefit from SoTA computer vision models.
>
> ---
>
>
> 2. > *The analysis of computation overhead is in section 3.3., but I could not find the cost of computing the low-rank projection matrix (p). It is neglectable, right?*
>
>    Yes, this is negligible. Specifically, the projection matrix is constructed only **once** before the training begins. Moreover, its creation time of 41us on a single-core CPU is negligible when compared to the overall training time.
>
>    $~$
>
>    Furthermore, in practice, generating and storing the WHT bases is **not required** for performing WHT. Similar to the Discrete Fourier Transform (DFT), representing the Walsh-Hadamard Transform (WHT) through matrix multiplication with a base matrix is needed only for the theoretical analysis. In practical applications, WHT is executed using the fast-WHT algorithm, an approach similar to calculating the FFT which does not require generating and storing the projection matrix.
>
> ---
>
>
> 3. > *Could you provide more experiments on vanilla vision transformers?*
>
>    We will provide such results in the final version of the paper, since running experiments, especially baseline methods, with a vanilla full-scale ViT would take over one week, given our resources; this is not feasible during the limited time of the rebuttal period.
>
>    $~$
>
>    It is worth noting that although EfficientFormer and EfficientFormerV2 used in our evaluations are CNN-ViT-hybrid models, the modules in their CNN part are actually **equivalent** to the feed-forward network in ViT, but implemented with pointwise convolution. Moreover, since the last stage of the large-scale models of EfficientFormer-L7 and EfficientFormerV2-L consist of ViT modules, the partial-training of these models is **similar** to training a vanilla vision transformer. Thus, we are confident that our experimental findings can be seamlessly transferred to other ViT models.
>
>    $~$
>
>    To address your concern regarding our method’s scalability and its compatibility with other training methods, we further test our method with the partial-training experimental setup for EfficientFormer-L1. Below, we discuss these results.
>
>    $~$
>
>    **Experimental results on large-scale datasets**
>
>    We test our method on a large-scale dataset Places365 [45], which contains over 1.8M training images and is more challenging than ImageNet (*i.e.*, models have a lower accuracy on Places365 than ImageNet). Below are our results for EfficientFormer-L1. Meaning of notations "LoRA", "LoRA-all" and $LP_{L_1}$-r is introduced in the general rebuttal section above [[link]](https://openreview.net/forum?id=NNtsO5L27J&noteId=8YCHi4VqDZ).
>
>    |Method|Speedup|MFLOPs|Accuracy [%]|
>    |-|-|-|-|
>    |Full BP|1.0x|1685.01| 55.30|
>    |LoRA|6.9x|242.61|50.64|
>    |LoRA-all|1.7x| 976.50|53.73|
>    |$LP_{L_1}-2$|7.2x|233.62|52.87|
>    |$LP_{L_1}-4$|3.5x|480.00|55.07|
>    |$LP_{L_1}-6$|2.1x|820.11|55.13|
>    |$LP_{L_1}-8$|1.2x|1397.02|55.39|
>
>    As shown, our method **scales well** on large scale datasets. For example, LBP-WHT with $LP_{L_1}$-2 base selection outperforms LoRA both in speed and accuracy; $LP_{L_1}$-8 has an even higher accuracy than the full-rank BP while achieving a 1.2x speedup.
>
>    $~$
>
>    **Experimental results when combining LBP-WHT with LoRA**
>
>    To show that our method is complementary to other existing methods, we combine our LBP-WHT with LoRA and show the results below.
>
>    |Method|Accuracy CF100 [%]|Accuracy CF10 [%]|Activation [MB]|Gradient [MB]|GFLOPs|
>    |-|-|-|-|-|-|
>    |Full BP|79.28|95.23|141|2352|121|
>    |LoRA-all|76.92|94.38|142|44|62|
>    |$LP_{L_1}-2$|75.61| 93.35|9|2352|8|
>    |$LP_{L_1}-4$|78.27| 94.60|29|2352|25|
>    |$LP_{L_1}-8$|79.34| 95.31|104|2352|91|
>    |$LP_{L_1}-2$+LoRA-all|73.27|92.62|9|44|4|
>    |$LP_{L_1}-4$+LoRA-all|75.48|93.74|29|44|13|
>    |$LP_{L_1}-8$+LoRA-all|76.58|94.33|104|44|48|
>
>    As shown, our method significantly reduces both the storage for the activation map (x) and the computational costs. On the other hand, LoRA efficiently reduces the memory usage needed to store the weights gradient. By combining both methods, we can **systematically reduce** both computation and memory costs, while maintaining the accuracy levels close to using LoRA alone. For instance, when combining LBP-WHT with $LP_{L_1}$-4 base selection and LoRA, we achieve a speedup of 4.7x and memory savings of 2.5x, with only a slight accuracy drop of 1.4% compared to employing LoRA alone. These results confirm the effectiveness of our method.

---

> > ### Comment · Reviewer_F5LH · 2023-08-16
> >
> > Thanks to the authors for providing their responses. I've read the rebuttal, and the authors have answered most of my questions.

---

### Author Rebuttal · Authors · 2023-08-09

We genuinely appreciate the reviewer's insightful comments and suggestions in reviewing our paper. Below we include additional references used in the rebuttal for all reviewers.

## Notation for Additional Experiments

We follow the same notation as the manuscript. To be more specific,

| Notation | Meaning |
|-|-|
| LoRA | Baseline method strictly following [14] which uses low-rank reparametrization solely in the ViT’s attention module. |
| LoRA-all | Baseline method applying [14] on all linear layers. |
|$LP_{L_1}$-r| Our LBP-WHT method with $LP_{L_1}$ base selection detailed in the Section 3.2. The "r" in $LP_{L_1}$-r is the hyper-parameter associated with $L_1$-based low-pass basis selection in Equation (8). As Figure 2(c) shows, each WHT basis is linked with a 2D coordinate (i, j). A larger i or j value correlates with a higher frequency change in one direction. The $LP_{L_1}$ selection method picks WHT bases with a smaller $L_1$ norm of the coordinate. For instance, $LP_{L_1}$-2 selects bases with the $L_1$ norm of the indices less than 2, that is, $|i| + |j| < 2$, such as (0, 0), (0, 1), and (1, 0). |


  $~$




## Additional References for Rebuttal
***These additional references are referred to throughout our responses to reviewer questions. References [1-44] are from the manuscript.***

[45] Zhou, B., et al. "Places: A 10 million image database for scene recognition." IEEE Trans. on Pattern Analysis and Machine Intelligence 40.6 (2017): 1452-1464.

[46] Dettmers, Tim, et al. "Qlora: Efficient finetuning of quantized llms." arXiv preprint arXiv:2305.14314 (2023).

[47] Kim, Jin-Hwa, et al. "Hadamard product for low-rank bilinear pooling." arXiv preprint arXiv:1610.04325 (2016).

[48] Li, Yanyu, et al. "Efficientformer: Vision transformers at mobilenet speed." Advances in Neural Information Processing Systems 35 (2022): 12934-12949.

[49] Li, Yanyu, et al. "Rethinking vision transformers for mobilenet size and speed." arXiv preprint arXiv:2212.08059 (2022).

[50] Kim, S., et al.. "Minimizing GPU kernel launch overhead in deep learning inference on mobile GPUs." Proc.Intl. Workshop on Mobile Computing Systems and Applications. 2021.

[51] Wang, Yiding, et al. "Egeria: Efficient DNN Training with Knowledge-Guided Layer Freezing." arXiv preprint arXiv:2201.06227 (2022).

[52] Lee, Jaejun, Raphael Tang, and Jimmy Lin. "What would elsa do? freezing layers during transformer fine-tuning." arXiv preprint arXiv:1911.03090 (2019).

---

> ### Author Response · Authors · 2023-08-16
> **Additional References for Author-Reviewer Discussion**
>
> [53] Chellapilla, Kumar, Sidd Puri, and Patrice Simard. "High performance convolutional neural networks for document processing." Tenth international workshop on frontiers in handwriting recognition. Suvisoft, 2006.

---

### Decision · Program_Chairs · 2023-09-21

**Decision:**

Accept (poster)

**Comment:**

This paper proposes a low-rank backpropagation algorithm based on Walsh-Hadamart Transformation that is specifically aimed at Vision Transformers (ViT). It demonstrates its effectiveness on pure ViTs and the hybrid ViT-CNN architecture, EfficientFormer. It is competitive with LoRA in terms of memory and FLOPs reduction.
Reviewers recognized that the paper addresses an essential problem with the transformer architecture with a novel solution that achieves promising results. Reviewers raised various points in their review, none of which the AC or SAC viewed as severe. The authors responded to these points, either clarifying or addressing them by additional experiments. The reviewers who responded to author feedback were satisfied by the response. The key recurring theme raised by multiple reviewers was the narrow applicability (claimed ViT-specific). The authors clarified that the approach was applicable to any backprop that employed matrix multiplication, including convolution. The use of EfficientFormer (hybrid ViT-CNN) is an example. Moreover, multiple vision tasks including classification and semantic segmentation were considered.
Less common points were brought up were:
- Insufficient literature review: the authors endeavoured to improve this and acknowledged some requested works were concurrent with the submission.
- Should add larger-scale datasets beyond ImageNet-1k: authors added additional experiments on Places365.
- Should use a SOTA baseline: authors clarified that the baseline was SOTA at the time of submission.
- Should do an open course release: author confirmed that code would be released with acceptance of the paper.
- The method's complementary to other approaches (e.g. LoRA) was claimed but not demonstrated: the authors demonstrated this by showing experiments that showed LBP-WHT with LoRA.
- Needs comparison with Gradient Compression: the authors explained that Gradient Compression was employed after backprop to improve communication while LBP-WHT focused on speeding up backprop.
- Should report GPU hours/memory savings: authors explained why FLOPs had been chosen as the unit of compute. The authors demonstrated memory savings.
- While backprop is sped up, doesn't address inference. This is effectively out-of-scope; the paper was clear about its focus.
The most negative reviewer did not respond to author feedback but it is plausable that they would have increased their score given that their negative points were directly addressed.
We implore the authors to take the promised changes into account in the camera-ready version of the paper and maintain their commitment to release code.